# RoboArm-NMP: a Learning Environment for Neural Motion Planning

## Abstract

We present *RoboArm-NMP*, a learning and evaluation environment that allows simple and thorough evaluations of Neural Motion Planning (NMP) algorithms, focused on robotic manipulators. Our Python-based environment provides baseline implementations for learning control policies (either supervised or reinforcement learning based), a simulator based on PyBullet, data of solved instances using a classical motion planning solver, various representation learning methods for encoding the obstacles, and a clean interface between the learning and planning frameworks. Using *RoboArm-NMP*, we compare several prominent NMP design points, and demonstrate that the best methods mostly succeed in generalizing to unseen goals in a scene with fixed obstacles, but have difficulty in generalizing to unseen obstacle configurations, suggesting focus points for future research.

## 1 Introduction

Fundamental in robotics, motion planning (MP) calculates a path for a robot to accomplish a task while avoiding obstacles (Latombe, 2012; LaValle, 2006). *Neural motion planning* (NMP, Qureshi et al. 2019) algorithms use a neural network (NN) to map a problem representation to a plan of actions, replacing the search-based methods of classical MP approaches with direct NN inference, and leveraging the NN's capability to recognize similarities between different problems to yield appropriately similar plans. Indeed, several NMP studies have recently shown promising results on various MP domains (Pfeiffer et al., 2017; Ichter & Pavone, 2019; Qureshi et al., 2019; Chiang et al., 2019; Jurgenson & Tamar, 2019; Ha et al., 2020; Strudel et al., 2021; Yamada et al., 2021; Liu et al., 2022; Fishman et al., 2022).

However, comparing the various different NMP techniques is difficult. Most previous studies focused on completely different problems (different robots, different obstacle configurations), with significant variations in the obstacle representations (e.g., are obstacles represented as a point cloud, image, or ground truth positions), and, most importantly, different post-processing methods that utilize either conventional MP techniques or other methods for transforming the NN output to a motion plan. Thus, at present, it is difficult to assess the fundamental capabilities of NMP, namely generalization to unseen goals or obstacle configurations, performance on 'difficult' instances such as narrow passages, and how to tease out the essential algorithmic ideas that make NMP work in general. To fill these gaps we propose *RoboArm-NMP*, a learning and evaluation environment for a 7 DoF robotic manipulator with a range of tasks of increasing difficulties.

Our goal in RoboArm-NMP is to lower the entry barrier for investigating NMP, by choosing a problem formulation and simulation environment that support both classical MP methods and learning approaches. RoboArm-NMP is written in Python, and builds on the free open-source Pybullet simulation environment (Coumans & Bai, 2016–2022), along with an easy to install code-base and pre-computed 80K trajectory demonstrations collected using classical motion planners (10K for each of our eight scenarios). Moreover, scenes from previous benchmarks (Chamzas et al., 2021) were added as test cases to allow comparisons to previous works, and additional scenes could be easily added on-demand. Finally, RoboArm-NMP includes implementations for several popular reinforcement learning (RL), imitation learning (IL), representation learning, and MP components, and a clean interface between them, facilitating both a thorough evaluation of existing NMP ideas, and simple development of new algorithms.

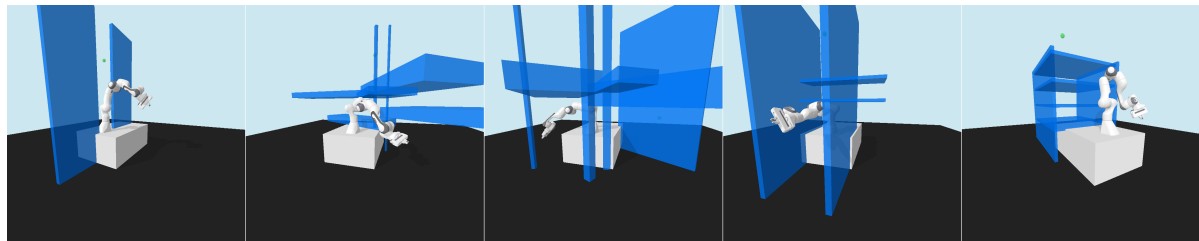

Figure 1: Example of *RoboArm-NMP* tasks: from the left, *double-walls* – a narrow gap scenario with fixed obstacles, two samples of *random boxes hard* demonstrating challenging narrow passages in our train data, and two test (OOD) tasks, *narrow shelves*, and *benchmaker bookshelf tall* (ported from Chamzas et al. 2021). See Section 4 for full tasks description.

Using RoboArm-NMP, we conduct a thorough investigation of several prominent algorithmic ideas in NMP, focusing on (1) the learning approach, i.e., IL vs. RL, and the importance of using hindsight experience replay (Andrychowicz et al., 2017) and relying on demonstrations; (2) generalization to unseen goals and obstacle configurations; and (3) how to best represent the goals and obstacles in the scene. We refrain from delving into post-processing techniques and, instead, concentrate on investigating the fundamental NN responsible for mapping observations to actions—the central element in every NMP algorithm. We posit that enhancing the learning of the NN poses a clearly defined question, and addressing it should enhance the performance of algorithms employing post-processing methods. Furthermore, our findings indicate that by centering our attention on this crucial component, we can cleanly compare various NMP approaches.

Our investigation findings can be summarised as follows: first, unlike previous works that utilized either demonstrations or hindsight methods, we find that the combination of both is essential – hindsight methods increase the positive signals encountered during training, accelerating learning in commonly visited parts of the state space, while demonstrations provide novel solutions for hard to reach goals. Second, the goal formulation either in configuration space, 3D position space, or a combination thereof matters, and creates noticeable performance gaps even when learning using the same algorithm. Finally, despite achieving near prefect success rate on domains with fixed obstacles, when the obstacles vary between episodes, and *generalization to obstacles* becomes a factor for success, performance of NMP policies drops considerably, making the best of these only slightly better than a heuristic policy that ignores the obstacles completely and navigates straight towards the goal. These findings may set a course for future NMP investigations.

## 2 Background and Problem Formulation

We present our problem formulation and provide relevant background material.

**Motion Planning:** In the motion planning (MP) problem (LaValle, 2006), an agent with configuration (joints) space $C$ is tasked with reaching a subset of goals denoted as $G \subseteq C$. The agent operates in a cluttered environment $E$ such that $F_E \subseteq C$ is the free space, and the collision predicate $Col_E : C \to \{1, 0\}$ s.t. $\forall c \in C : c \in F_E \Leftrightarrow Col_E(c) = 0$ defines collisions with obstacles. A *feasible* solution to a MP problem is a mapping $\tau : t \in [0, 1] \to C$ that represents a continuous sequence from the initial configuration $c_0 = \tau(0) \in C$ to a goal configuration $\tau(1) = g \in G$, s.t. $\forall t \in [0, 1] : Col_E(\tau(t)) = 0$, i.e. the entire path, $\tau$, is in the free space. It is often desired that $\tau$ is optimal with respect to some cost function, such as minimal time to goal, minimal inverse clearance, etc. Formally, let $f_E(\tau|G) \in R$ be a scalar function, we are most interested in the feasible solution $\tau^* = \arg\min_\tau f_E(\tau|G)$.

A popular approach for solving MP is to use sampling-based motion planners (SBMP; Kavraki et al. 1994; LaValle & Kuffner Jr 2001), that create a *roadmap* – a discrete approximation of $C$ as a graph. The roadmap nodes correspond to configurations and edges represent primitive motions between two adjacent configurations. The input for SBMP is a query $(s, g)$, $s, g \in C$ denote the start and goal, both are added to the roadmap. Then the SBMP finds a solution by searching for a path in the roadmap that starts in $s$ and ends in $g$, and expands the graph by some expansion method if required.

**Motion Planning Formulation:** in MP the agent is required to reach various goals from various starting states, therefore we formulate MP as a goal-conditioned Markov decision process (GC-MDP Bertsekas 1995; Andrychowicz et al. 2017). The GC-MDP formulation is equivalent to a discrete-time version of the standard MP formulation (LaValle, 2006), and unifies both classical MP approaches and RL/IL approaches. A GC-MDP is a tuple $M = \langle S, G, A, \rho_0, P, C, T \rangle$, where the agent's continuous state, goal, and action spaces are $S$, $G$, and $A$. At the start of each episode, the initial state and goal are sampled from a joint distribution $s_0, g \sim \rho_0$. Later, at every discrete timestep $t \in [0 \ldots T-1]$, a policy $\pi_t : S \times G \to \Delta(A)$ predicts the next action, and a new state is sampled from the environment according to the transition function $P : S \times A \to \Delta S$; subsequently, a scalar cost is incurred according to the cost function $C : S \times A \times S \times G \to \mathbb{R}$. The process repeats and terminates after $T$ timesteps. The objective is to minimize the expected cumulative cost $J^\pi = \mathbb{E}_{\rho_0, P, \pi} \left[ \sum_{t=0}^{T-1} C(s_t, a_t, s_{t+1}, g) \right]$. In the case of NMP, the state space $S$ is composed of the positions of the joints, as well as the end-effector position. The actions $A$ are joint positions displacement, and the goal region $G$ is a ball in $\mathbf{R}^3$ around some end-effector position. The transition function $P$ is deterministic and follows the dynamics of the robot (according to the displacement specified by the action). Finally, $C$ is 0 when the agent first reaches $G$, otherwise it is -1[1].

**Behavioral Cloning (BC):** a simple method to train an NMP policy is to imitate an SBMP planner (Pfeiffer et al., 2017; Qureshi et al., 2019; Yamada et al., 2021; Liu et al., 2022; Fishman et al., 2022). Given SBMP-produced trajectories, a policy is learned to maximize the (log) likelihood of the observed action given the current state and goal (SBMP algorithms produce a state trajectory of the form $\tau = \{s_0, s_1 \ldots, s_l = g\}$. To obtain actions, we exploit the fact that actions are defined as state displacements, and obtain an action trajectory $\{s_1 - s_0, s_2 - s_1 \ldots s_l - s_{l-1}\}$. For more details see the "data collection" Section D.) .

**Demonstration-guided off-policy RL:** Most RL algorithms use random noise heuristics for data collection. However, these heuristics are not effective for long-horizon sparse reward tasks like MP, where robots must navigate through 'long corridors' in the state space (e.g. Figure 1 second scenario from the right). To address this, following previous works (Jurgenson & Tamar, 2019; Ha et al., 2020), we incorporate demonstrations of successful plans into the data, and use off-policy RL algorithms, which can learn from data collected by an agent that is not the RL policy. We use TD3 (Fujimoto et al., 2018) and SAC (Haarnoja et al., 2018), two popular off-policy RL algorithms for continuous control, widely used in NMP (Yamada et al., 2021; Liu et al., 2022; Strudel et al., 2021; Jurgenson & Tamar, 2019; Ha et al., 2020). The demonstrations are created using SBMP from the same initial state. Importantly, these demonstrations can be computed in advance for all possible starting states, which speeds up learning compared to methods that add expert labels during learning such as DAgger (Ross et al., 2011). As off-policy data can lead to training instability (Fujimoto et al., 2019), we inject demonstrations only upon failure with a certain probability (Jurgenson & Tamar, 2019; Ha et al., 2020). This technique is referred to as "demo-injection" and algorithms using this technique are denoted with an "-MP" suffix (e.g., SAC-MP) in accordance with the term "motion-plan".

**Hindsight learning in goal-conditioned tasks:** To handle sparse rewards in GC-RL, hindsight (Andrychowicz et al., 2017) is commonly used by re-interpreting a failed trajectory as successful. For this, an *imagined* goal is selected such that the original (failed) trajectory accomplishes this goal (For instance, the imagined goal could be the last state in the failed trajectory). Then rewards are recomputed based on the imagined goal, and this relabelled data is added to the off-policy algorithm.

**Learning obstacle representations:** To generalize to varying obstacle configurations, the RL policy must have some sensing of the scene. One such solution is encoding a sensor-observation (point cloud, or images) from multiple viewpoints in the scene into a unified latent vector $z_E$, and augment the state space with that vector ($z_E$ is fixed throughout the episode). In our work, we encode images with *VAE* (Kingma & Welling, 2013) and *VQ-VAE* (Van Den Oord et al., 2017), and point clouds with PointNet++ (Qi et al., 2017b).

---

[1]Other NMP cost functions are possible, and can be implemented in our environment, see Appendix B

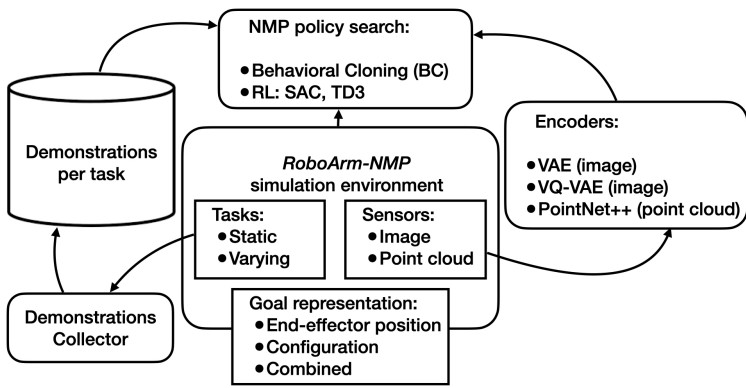

Figure 2: Schematic diagram of the *RoboArm-NMP* components.

## 3   Related work

While several previous NMP studies demonstrated successful applications of neural networks to various modules in the MP pipeline (Ichter & Pavone, 2019; Qureshi et al., 2019; Chiang et al., 2019; Pfeiffer et al., 2017; Yamada et al., 2021; Liu et al., 2022; Strudel et al., 2021; Jurgenson & Tamar, 2019; Ha et al., 2020; Fishman et al., 2022), there is not yet a framework for studying, developing, and evaluating NMP algorithms in a common setting, which is the core contribution of this work. One evaluation challenge is that several works add various post-processing steps (Qureshi et al., 2019; Yamada et al., 2021; Liu et al., 2022), which is important for improving final performance, but obfuscates the evaluation of NMP's most basic ingredient – the policy (Strudel et al., 2021; Jurgenson & Tamar, 2019; Ha et al., 2020). The challenge in evaluation is exacerbated as different works investigated different tasks (MP + manipulation; Yamada et al. 2021; Liu et al. 2022 vs. only MP; Strudel et al. 2021; Jurgenson & Tamar 2019 vs. navigation; Pfeiffer et al. 2017; Chiang et al. 2019), different robots (e.g., point robot; Strudel et al. 2021 vs. multiple robotic arms; Ha et al. 2020), and were written in different software frameworks (ROS in Pfeiffer et al. 2017 vs. Pybullet in Ha et al. 2020), adding a technical difficulty. In this work we compare the learning components suggested in previous works under a single environment, with the goal of teasing out the most important algorithmic design choices.

A major NMP challenge is generalization to held-out obstacle configurations in very cluttered environments. In Jurgenson & Tamar (2019), only 2D scenes were tested, while other works predicted actions from state vectors directly (Yamada et al., 2021; Ha et al., 2020) (requiring perfect knowledge of the environment, which is an unrealistic assumption for unstructured scenarios), had workspaces with only a few small obstacles (Yamada et al., 2021; Liu et al., 2022; Ha et al., 2020), or considered tasks where the obstacles did not limit the robot's range of motion much (Fishman et al., 2022) making collisions easy to avoid. Thus, while these works demonstrated impressive results, it is difficult to assess whether these results scale to harder scenarios, and to identify the real challenges in the field.[2] Our environment contains tightly-packed 3D scenes (see Figure 1), and we focus on comparing different obstacle encoding schemes, which to our knowledge has not been reported previously. Importantly, our unified environment allows to draw clear challenges for the current state of the art, which were not evident from previous studies.

For classical MP, C++ based environments Şucan et al. (2012) and benchmarks Moll et al. (2015); Chamzas et al. (2021) have been developed. However, these environments are difficult to integrate with deep-learning software, which is mostly developed in Python (e.g., Pytorch Paszke et al. 2019). More importantly, *non of these environments contained enough data* for training deep learning-based policies, or a simulator for RL based methods. In *RoboArm-NMP* we included 10K motion plans for 8 scenarios, totaling in 80K trajectories. We point out, however, that some MP benchmarks such as Chamzas et al. (2021) contain *tasks* that are

---

[2]An extended technical comparison with selected prior work appears in Appendix C.

relevant for our robotics arm scenario. To build on this, we add tasks from Chamzas et al. (2021) as test scenes in RoboArm-NMP, thereby bridging between the evaluation of SBMP methods and NMP.

## 4 Environment description

We propose *RoboArm-NMP*, a NMP simulation, learning and evaluation environment for a 7 DoF robotic arm. RoboArm-NMP is comprised of (see Figure 2):

- A simulation environment based on Pybullet, with configurable properties relevant to MP such as goal tolerance, and collision sensitivity (see appendix Section B for more details).
- Visual sensor implementations for images and point clouds. Sensors are modular, stack-able, and configurable allowing the user to mix and match between sensor types.
- Tasks: a collection of tasks, with either fixed obstacles, or with obstacles in varying positions and shapes[3]. Additionally, we define a set of test tasks – tasks with conceptually different obstacles from the training tasks, to evaluate generalization to unseen obstacles.
- Data: A set of 10K test cases per-task for evaluations. A data-set of 10k demonstrations (from a SBMP) per-task, except for the test tasks (see below for detailed description).
- Baseline learning methods capable of incorporating demonstrations: both behavioral-cloning, and off-policy learning algorithms (TD3; Fujimoto et al. 2018, SAC; Haarnoja et al. 2018, HER; Andrychowicz et al. 2017, and demo-injections; Jurgenson & Tamar 2019; Ha et al. 2020). These algorithms are common in previous NMP works (Yamada et al., 2021; Liu et al., 2022; Strudel et al., 2021; Jurgenson & Tamar, 2019; Ha et al., 2020).
- A set of NN encoders (VAE; Kingma & Welling 2013, VQ-VAE; Van Den Oord et al. 2017, and PointNet++; Qi et al. 2017b) used in previous works (Ichter & Pavone, 2019; Pfeiffer et al., 2017; Qureshi et al., 2019; Strudel et al., 2021; Fishman et al., 2022) that map sensor readings into representation vectors suitable for further processing by NMP policies (code and weights of trained models).
- A simple interface between SBMP components and ML algorithms, allowing for additional demonstrations to be collected and verified.

Thus, RoboArm-NMP provides a complete environment to explore both the algorithmic aspects (i.e., path planning) and representation learning aspects (i.e., how to encode obstacles) of NMP.

We next describe the different types of tasks in RoboArm-NMP, which were designed to measure an important property of the NMP policy – generalization. Generalization in NMP can be classified to either generalization to unseen start-goal pairs in the same obstacle configuration, or to unseen obstacle configurations. We thus suggest two sets of tasks, `goal-generalization tasks` that only test start-goal generalization, and `obstacle-generalization tasks` that test for both types.

**Goal-generalization tasks:** We defined five tasks with fixed obstacles. *No obstacles*, contains the robot, the table and the floor [4], and provides an easy proof-of-concept for algorithmic ideas. Next, *wall*, *double wall wide gap*, and *double walls* demonstrate with increasing difficulty the narrow passage problem[5]. Finally, *boxes* is a task with less structure in the obstacles placement.

**Obstacle-generalization tasks**: We created tasks where obstacles in the form of shelves, poles, and walls are sampled from a fixed distribution around the robot. The agent perceives the obstacles using images or point-clouds from four sensors located in cardinal directions looking at the robot (see Figure 10), thus successfully reaching the goal also requires 3D visual understanding of the agent. The `obstacle-generalization`

---

[3]Obstacles do not change shape and position between episodes, the change occurs in between episodes.

[4]Collision can happen not only between the robot and the table / floor, but also self-collisions between different links of the robot itself.

[5]The *narrow passage problem* Jurgenson & Tamar (2019) refers to narrow gaps between obstacles that the agent must traverse to reach the goal.

`tasks` include *random boxes easy*, *random boxes medium*, and *random boxes hard*, corresponding to a sample of 2-4, 3-6, and 4-8 boxes, respectively (see Figures 4, 5, and 6).

Ideally, a robot trained on a variety of procedurally generated obstacle configurations should generalize to common practical scenarios. To investigate this hypothesis, we set aside a curated set of obstacle configurations representing particular arrangements of interest (e.g., "shelves") – the `OOD tasks` (out-of-distribution tasks).

`OOD tasks`: We defined three fixed environments; *narrow shelves*, *three shelves*, and *pole shelves*, and ported two environments from Chamzas et al. (2021) (*benchmaker bookshelf tall* and *benchmaker bookshelf thin*). These scenarios are markedly different from scenarios sampled as described above, and we do not allow training on them. Figure 1 shows two queries from the training task we seek to generalize from – *random boxes hard* domain (second and third images from the left), and the test only tasks of *narrow shelves*, and *benchmaker bookshelf tall* (fourth and fifth from the left).

Compared with other MP learning environments such as Chamzas et al. (2021), we prioritize workspaces with cluttered obstacles, that create narrow passages[6]. Furthermore, our challenging task design allows testing both the *generalization capabilities* of NMP algorithms to similar yet different obstacle distributions, as well as investigate various *obstacle representations*, a crucial gap in our understanding of NMP algorithms.

## 5 Experiments

Using RoboArm-NMP, we now investigate common solutions of the NMP problem. We isolate the contribution of every algorithmic component, and build towards a clear picture that identifies current NMP challenges. Specifically, we investigate the following aspects of NMP:

1. What are crucial algorithmic components in NMP (i.e. problem formulation, prior knowledge, learning method, etc'.)

2. We also investigate how well do NMP algorithms generalize in two contexts, first when the train and test obstacles distribution are identical, and then when they are different.

3. Finally, we investigate the inference speed of NMP algorithms, namely, the prediction speed of the NN.

We start by describing our experiments setup (Section 5.1), then in Sections 5.2 and 5.3 we investigate the algorithmic components (question (1) above). In Section 5.4 we investigate the generalization capabilities of question (2). Finally, in Section 5.5 we investigate inference times of the final main question.

### 5.1 Setup

**Data and evaluation set:** We use the tasks described in Section 4. For each task we sampled 10K $(s, g)$ queries as the test set, as well as 10K trajectories using different queries (for details regarding demonstrations collection see the appendix Section D). In our results we also denote the success rate of a simple `go-to-goal` policy that moves directly to the goal, while completely ignoring the obstacles[7]. This policy shows how likely it is that a single motion towards the goal is enough to reach the goal, and thus measures the "hardness" of each task.

**Model training and inference:** For training BC agents we train for 1M batches (for which we observed convergence of all agents), and for the RL agents we train for 1M and 7M environment steps for goal-generalization and obstacle-generalization tasks correspondingly. Our `goal-generalization` networks have two hidden feed-forward layers with 256 neurons each (common in recent RL and NMP works; Jurgenson & Tamar 2019; Huang et al. 2022). For the `obstacle-generalization` tasks we use four layers (see Section 5.3

---

[6]Known to be hard for NMP algorithms (Jurgenson & Tamar, 2019).

[7]The `go-to-goal` policy gets as input the current and goal configurations $c$ and $c_g$, and takes an action in the direction $a_t = c_g - c$.

for details). We searched for hyper-parameters for all three algorithms (BC, SAC, TD3); parameter sweeps were performed on the `goal-generalization` tasks, and the best values found were used in the rest of the empirical evaluation. These sweeps did not include hyper-parameters under investigation in this section (i.e., *demo-injection p*) – for full details see section I in the appendix. To choose a model checkpoint we used a small, fixed validation set of queries. We repeat each experiment four times and report means and standard deviations.

**Experiments pipeline:** To cleanly investigate how to best represent a scene for NMP, we separate between scene representation learning and policy optimization. First, we train a visual encoder to extract a latent obstacle representation from the observations, $z_E$. Then, we train a policy conditioned on this latent representation, namely: $\pi(a|s, g, z_E)$. This allows us to probe the understanding of a 3D representation in isolation from the policy optimization process. We note that users of *RoboArm-NMP* are not required to follow this pipeline, and can opt for an end-to-end approach instead.

**Learning the visual encoders:** In this work we learn visual encoders with a supervised or an unsupervised objective, independent of the rewards. We follow previous works and provide support for perception of point clouds (Strudel et al., 2021; Qureshi et al., 2019; Fishman et al., 2022) and images

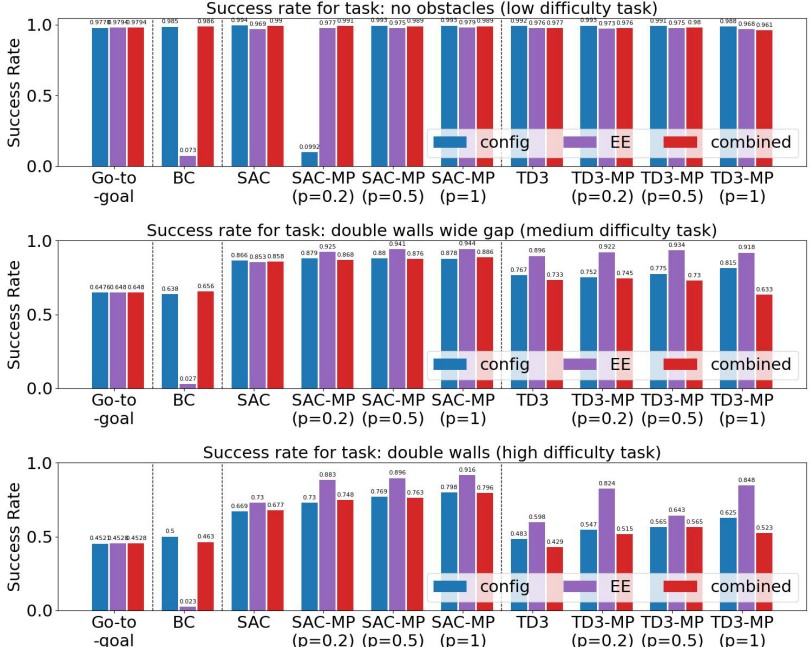

Figure 3: we compare the success rate of different algorithms (X axis) with different goal representations (colors). (a) *No obstacles task* (b) *Double walls wide gap task* (c) *Double walls task*.

(Ichter & Pavone, 2019; Jurgenson & Tamar, 2019), and learn visual representations based on commonly used encoders such as PointNet++ (Qi et al., 2017b), VAE (Kingma & Welling, 2013), and VQ-VAE (Van Den Oord et al., 2017). For full technical details, including the processing of sensor readings, see Section E in the supplementary material.

**Learning policies:** Policies are trained with the TD3 (Fujimoto et al., 2018) or SAC (Haarnoja et al., 2018) RL algorithms, or with BC as explained in Section 2. For the *goal-generalization tasks* the input of the policy is $(s, g)$ the current robot state and the goal representation, and since obstacles change between episodes in the *obstacle-generalization tasks*, we add the latent vector $z_E$ from the visual encoder – $(s, g, z_E)$.

## 5.2 Results for `goal-generalization tasks`

We begin our investigation into the algorithmic components of NMP training (following question (1)), using the `goal-generalization tasks`.

**Goal representation matters:** We start with the question of how to best represent goals in NMP. A useful specification is a task-space goal, i.e., the goal pose of the end effector (EE). Unfortunately, we discovered that this is not ideal for all algorithms (see analysis below). Instead we suggest two alternative goal formulations, the goal *configuration* of the robot, and a combined representation where both the end-

|  | Hindsight | No Hindsight |
|---|---|---|
| SAC (No demos) | $0.990 \pm 0.002$ | $0.001 \pm 0.000$ |
| SAC-MP $p = 0.2$ | $0.991 \pm 0.002$ | $0.017 \pm 0.004$ |
| SAC-MP $p = 0.5$ | $0.989 \pm 0.002$ | $0.253 \pm 0.041$ |
| SAC-MP $p = 1.0$ | $0.989 \pm 0.002$ | $0.599 \pm 0.027$ |

Table 1: Exploration solutions: SAC with *combined* goal representation in the *no-obstacles*. We investigate combinations: hindsight (on / off), demo-injection ($p \in 0, 0.2, 0.5, 1$)
.

|  | random boxes hard (trained) | random boxes medium | random boxes easy | three shelves | pole shelves | narrow shelves |
|---|---|---|---|---|---|---|
| VQ-VAE (no-demos) | $0.304\pm0.007$ | $0.380\pm0.009$ | $0.483\pm0.012$ | $0.792\pm0.009$ | $0.721\pm0.020$ | $0.493\pm0.026$ |
| VQ-VAE $p = 0.5$ | $0.302\pm0.008$ | $0.388\pm0.012$ | $0.495\pm0.017$ | $0.816\pm0.015$ | $0.741\pm0.024$ | $0.497\pm0.043$ |
| VQ-VAE $p = 1.0$ | $0.274\pm0.009$ | $0.347\pm0.011$ | $0.445\pm0.014$ | $0.750\pm0.028$ | $0.696\pm0.015$ | $0.405\pm0.022$ |
| Go-to-goal | $0.2891$ | $0.3697$ | $0.4914$ | $0.8316$ | $0.7207$ | $0.5393$ |

Table 2: Out of distribution performance: we compare our VQ-VAE models trained only on *random boxes hard* on (1) two easier yet similar tasks *random boxes medium*, and *random boxes easy*, and (2) on three hand-crafted test tasks *three shelves*, *pole shelves*, and *narrow shelves*.

effector's goal location and goal configuration are provided[8]. We denote all options as *EE*, *config*, and *combined*. Focusing on the results in Figure 3, we can see that clearly for BC policies, goal representation matters, and the "natural" way to encode a goal using the EE alone is difficult for the agent. On the other hand, both RL algorithms (SAC and TD3) were able to get decent results for all goal representations, but clearly, the *EE* goal representation obtains the best results. Interestingly, the *combined* representation, although being a super-set of the information in either *config* and *EE* is not ideal for RL. This is in contrast to the BC policy where the *combined* is on-par or even slightly better than the *config* goal representation.

We explain these results as follows, the *config* representation creates less ambiguity for BC algorithms when following a trajectory as the same EE position may represent different joint configurations. However, for the RL agent, the EE representation allows more freedom, since the agent can obtain reward from any of the configurations that reach the required EE position.

**Handling sparse rewards:** *Hindsight* and *demo-injections* (see Section 2) are two powerful tools to overcome the sparse rewards problem in NMP. However, how to best exploit demonstrations such that they 'play well' with the RL algorithm is still an open question (Jurgenson & Tamar, 2019; Ha et al., 2020). In this experiment we compare both approaches in the simplest *no-obstacles* task; the results are shown in Table 1. For hindsight, we follow the best-practice and set the probability for hindsight to 0.8 (see Section 2). For demo-injection we investigate several values for $p$.

From Table 1 we observe that RL methods fail without demonstrations or hindsight in this relatively easy task, but introducing hindsight clearly boosts the success rate. While here demo-injection does not improve over hindsight, by observing results in other goal-generalization tasks, such as double walls, it is clear that tasks with narrow passages benefit more from the demo-injections.

We conclude that, contrary to previous works that claimed instability issues of HER in NMP (Jurgenson & Tamar, 2019), HER is crucial for learning effectively. And although HER by itself is not effective for narrow passages, it can be combined with demo-injections to gain from the best of both worlds.

**Comparison of policy optimization algorithms:** We next compare different NMP policy search methods in the goal-generalization tasks. Based on our earlier results, we apply both hindsight and demo-

---

[8]In NMP applications, the *config* could be obtained in several ways, such as inverse kinematics, or a learned mapping from EE position to configurations.

|  | VAE | VQ-VAE | PointNet++ |
|---|---|---|---|
| BC | $0.047 \pm 0.032$ | $0.150 \pm 0.045$ | $0.177 \pm 0.020$ |
| SAC (no demos) | $0.284 \pm 0.003$ | $0.304 \pm 0.007$ | $0.279 \pm 0.007$ |
| SAC-MP $p = 0.5$ | $0.234 \pm 0.024$ | $0.302 \pm 0.008$ | $0.254 \pm 0.032$ |
| SAC-MP $p = 1.0$ | $0.199 \pm 0.012$ | $0.274 \pm 0.009$ | $0.240 \pm 0.016$ |
| Go-to-goal | 0.2891 | | |

Table 3: obstacle-generalization tasks: we compare different encoders (VAE, VQ-VAE, PointNet++), with different policy learning algorithms (BC, SAC+HER), and different probabilities to add demos (0., 0.5, 1.) when using the SAC algorithm. All policies in this experiments have 4 hidden-layers.

injection. Comparing both SAC and TD3 to BC shows that the RL methods are clearly superior[9]; all RL variants obtain higher success rates then pure BC in all but the *no-obstacles* task, for which all methods are already almost perfect. In agreement with previous works (Jurgenson & Tamar, 2019), which hypothesized that RL provides crucial data around obstacle boundaries compared to BC, we conclude that RL indeed provides substantial benefit for NMP.

### 5.3 Results for `obstacle-generalization tasks`

The objective of the experiments in the `obstacle-generalization tasks`, are to understand the benefits of different visual encoders (question (1)), and to investigate generalization to different obstacle configurations (question (2)). To limit the number of parameters to investigate, we make some use of the conclusions from the `goal-generalization` experiments, and choose the *combined* goal representation, HER, and SAC for our RL experiments. However, as our experiments with demo-injection did not yield a conclusive preference, we will investigate its use further. For the following experiments we use a deeper network than in `goal-generalization tasks`, as the network must now also process $z_E$ – the encoding of the scene. Specifically, we increase the depth of $\pi$ from two to four hidden layers.

**Which visual encoder to use:** We next compare three visual encoders architectures: VAE (Kingma & Welling, 2013), VQ-VAE (Van Den Oord et al., 2017), and PointNet++ (Qi et al., 2017b); results are shown in Table 3.

First, regarding policy optimization, we observe that as in the `goal-generalization tasks`, BC is inferior to RL (SAC+HER in our case) as the success rate of BC for each model is lower than all of its RL counterparts. Next, when we compare the success rates of the various SAC models, we see that contrary to the `goal-generalization tasks`, here, adding demonstrations decreases the performance of the policy substantially (the success rate of SAC is greater than SAC-MP with $p = 0.5$, which in turn is greater than SAC-MP with $p = 0.5$ for every visual encoder). We hypothesize that in `obstacle-generalization tasks`, the policy makes more errors as it is more difficult to take the high-dimensional context into account, which aggravates the distribution shift problem, adversely affecting off-policy learning, and thereby reducing the utility of demonstrations. Finally, we see that VQ-VAE performs better than both VAE and PointNet++. Similar to conclusions from recent deep RL works (Hafner et al., 2020), we hypothesize that this performance gap is due to the discrete nature of $z_E$ when using a VQ-VAE.

However, to our surprise we found that *the `go-to-goal` baseline policy, which disregards obstacles, outperforms nearly all NMP methods*! The only exception is the VQ-VAE based approach, which has a slight but statistically significant advantage. This entails that although some aspects of the problems were indeed learned (as indicated the positive success rates and by the increasing rewards during training), most model are roughly equivalent or dominated by a simple scripted behavior.

---

[9]See Tables 5, 6, and 7 for full results in the `goal-generalization tasks`.

### 5.4 Results for `OOD tasks`

We now investigate how well policies trained in the *random boxes hard* perform in tasks with different obstacles distributions, see Table 2. We evaluate on the *random boxes medium* and *random boxes easy*, where the sampling of obstacles is the same but their count is lower than *random boxes hard* (full analysis in Section J). Then, we evaluate on `OOD tasks`, where obstacles are not sampled but instead represent common industrial settings inspired by previous NMP works (Chamzas et al., 2021). We focus here on the VQ-VAE policies as these are the ones with the best performance in *random boxes hard* that were able to exceed the `go-to-goal` baseline (for full results see Table 8 in Section J).

**The `OOD tasks`**: We observed that SAC-MP ($p = 0.5$) policies outperformed the `go-to-goal` policy in `pole shelves` tasks and came close to matching it in other tasks. It is noteworthy that the generalization experiments in this section revealed a distinct impact of demo-injection. While demo-injection negatively affected performance in `random boxes hard`, with $p = 0.5$ it exhibited superior success rates for all other tasks. Currently, we lack an explanation for this phenomenon and propose it as a compelling avenue for future research, especially considering the positive influence of demo-injection in goal-generalization tasks.

Finally, noticing the high success rates in all three `OOD tasks`, we reach an interesting take-away: without a proper baseline to frame the results (in our case `go-to-goal`), success rates alone are not enough to evaluate NMP policies and must be taken with a grain of salt.

### 5.5 How fast are NNP solutions?

To effectively use NMP on a real systems policy inference must be quick, thus we investigate the inference time of the `OOD tasks` policies (question (3)). We find that our policies can predict *an entire trajectory* in less than 0.25 seconds on average. Moreover, our inquiry suggests that if our policies are used in scenes where the obstacles move during the episode, we can approach real time predictions with inferences around 100Hz. See appendix Section F for details.

## 6 Limitations

**Post-processing methods:** In this work we focused on learning a NMP policy as the basis of our investigation. However, a complete deployment of a NMP solution requires other software components that ensure successful execution and hardware safety. Post-processing methods (Kavraki et al., 1994; LaValle & Kuffner Jr, 2001) are a popular choice, and are orthogonal to the NMP policy. We hypothesize, that improvements of the NMP policy would reduce the time spent planning thus making the overall system better (Qureshi et al., 2019; Yamada et al., 2021; Liu et al., 2022). Metrics and benchmarks comparing the integration of the two are not covered by this work and would require further investigation.

**Other robotic settings:** we investigated NMP for robotic arms, but other systems, such as drones (Hanover et al., 2023), cars, and quadrupeds (Tsounis et al., 2020), may present fundamentally different challenges, which could be interesting to explore. For instance, drones operating in outdoor settings should account for stochastic dynamics due to air currents, which presents a different control problems from the relatively deterministic robotic arm setting. Different from both settings, quadrupeds require frequent contacts with the environment in order to move, thus requiring a different definition for collisions. It would be interesting to see if our findings (e.g. the role of demonstrations, HER etc'.) also extend to these systems.

**Task and motion-planning (TAMP):** MP for robotic arms is commonly used as a sub-component in a larger planning problem such as TAMP (Kaelbling & Lozano-Pérez, 2011). In this work we focus only on the basic MP component emphasizing collision-avoidance. It would be interesting to see if NMP improvements could increase the overall performance of manipulation tasks with cluttered workspaces as well.

**Real-world applicability:** RoboArm-NMP is a simulated environment, as such, it doesn't deal with the simulation-to-real gaps for transferring the trajectories found into the real world, particularly in sensor noise and in dynamics (i.e., precise tracking of the NMP trajectory). Of the two, we hypothesize that sensor noise will be more challenging, as tracking control is well established (Haddadin et al., 2022). However,

this investigation would require creating an environment, similar to RoboArm-NMP, on a real-world setup. Challenges for this setup include time-consuming demonstrations collection, success and collision detection (for reward functions and large scale evaluations), task and obstacle design, and more. We recommend focusing in simulated environments at present as they are easier to control and faster to experiment on, but as our experiments clearly demonstrate, still provide a significant challenge to modern NMP algorithms.

## 7 Outlook and discussion

We presented RoboArm-NMP, an evaluation environment for the emerging field of NMP, and a one-stop-shop for infrastructure requirements for the NMP practitioner. RoboArm-NMP includes configurable environments, learning algorithms, data, pre-trained models, and code to collect and use demonstrations. RoboArm-NMP also provides a common set of tasks to asses progress in NMP, and to help investigate various components frequently used in NMP research.

In our experiments we focused on the fundamental building block of NMP – the NN policy, and investigated aspects for training it, such as obstacle-configurations, goal-representations, and optimization frameworks. However, our most interesting findings relate to tasks where the agent must infer the obstacles and react accordingly, based on a vector encoding of the scene. We investigated encoders commonly used in previous works (Ichter & Pavone, 2019; Pfeiffer et al., 2017; Qureshi et al., 2019; Strudel et al., 2021; Jurgenson & Tamar, 2019), and found that in contrast to impressive performance on fixed obstacle tasks, all of the NMP methods we evaluated were at best on-par with a simple heuristic baseline. This result hints that generalizing to obstacle configurations unseen during training is still an unsolved problem, and we believe that NNs such as Transformers (Vaswani et al., 2017), residual networks (He et al., 2016) or NERFs (Mildenhall et al., 2021) may better capture and utilize information in the 3D scene – an interesting direction for future research.

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

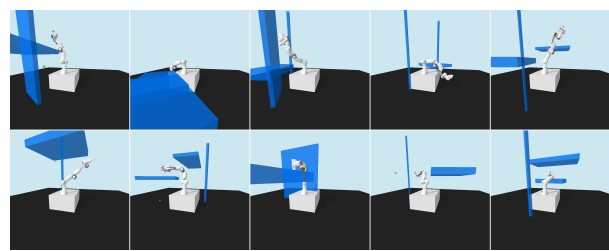

Figure 4: *Random boxes easy* example queries. The start configuration, goal state and obstacles (2-4) are sampled randomly. The robot is at the starting state and the green sphere represents the end-effector goal position.

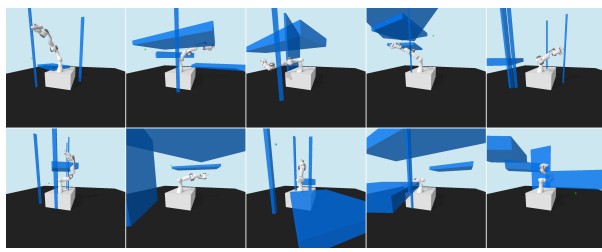

Figure 5: *Random boxes medium* example queries. The start configuration, goal state and obstacles (3-6) are sampled randomly. The robot is at the starting state and the green sphere represents the end-effector goal position.

# A    Motion Planning Problem

In the motion planning (MP) problem (LaValle, 2006), an agent with configuration (joints) space $C$ is tasked with reaching a subset of goals denoted as $G \subseteq C$. The agent operates in a cluttered environment $E$ such that $F_E \subseteq C$ is the free space, and the collision predicate $Col_E : C \to \{1, 0\}$ s.t. $\forall c \in C : c \in F_E \Leftrightarrow Col_E(c) = 0$ defines collisions with obstacles. A *feasible* solution to a MP problem is a mapping $\tau : t \in [0, 1] \to C$ that represents a continuous sequence from the initial configuration $c_0 = \tau(0) \in C$ to a goal configuration $\tau(1) = g \in G$, s.t. $\forall t \in [0, 1] : Col_E(\tau(t)) = 0$, i.e. the entire path, $\tau$, is in the free space. It is often desired that $\tau$ is optimal with respect to some cost function, such as minimal time to goal, minimal inverse clearance, etc. Formally, let $f_E(\tau|G) \in R$ be a scalar function, we are most interested in the feasible solution $\tau^* = \arg\min_\tau f_E(\tau|G)$.

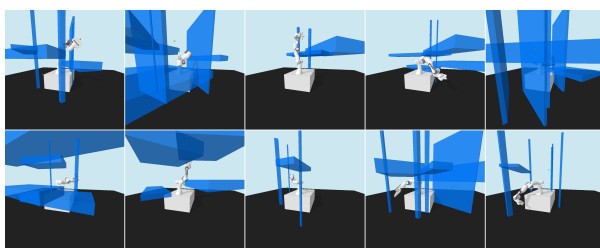

Figure 6: *Random boxes hard* example queries. The start configuration, goal state and obstacles (4-8) are sampled randomly. The robot is at the starting state and the green sphere represents the end-effector goal position.

# B   Extended NMP goal-conditioned MDP description

We briefly reiterate the MDP formulation for NMP as a goal-conditioned task (from Section 2), we expand on all the options available in *RoboArm-NMP*, not just the ones used to evaluate algorithms. As a reminder a finite-horizon GC-MDP is defined as a tuple $M = \langle S, G, A, \rho_0, P, C, T \rangle$, and in the case of NMP we define:

1. The state space $S$ is composed of the positions and velocities of the joints, as well as the end-effector position. The positions are normalized to be between $[-1, 1]$ to allow the NNs to better handle the different scales. If $m, M \in \mathbf{R}^7$ are the minimal and maximal joint configuration (7 being the DoF of the robot), then the **normalized configuration space** is $[-1, 1]^7$ and the linear transformation from configuration $c \in \mathbf{R}^7$ to a normalized configuration $s \in [-1, 1]^7$ is

$$s = \frac{2c - (M + m)}{M - m}$$

.

2. The actions $A$ in the experiments are *relative*, i.e. these are differences $\Delta s \in [-1, 1]^7$, that are combined with the current state (in normalized configuration form) $s$, to create the next state $s' = s + \Delta s$. The other option supported by *RoboArm-NMP* is for a *subgoal*, i.e. the value is the next normalized configuration the robot should move towards (simply $s'$). Either of the above options determine the next target normalized configuration, $s'$, and then we apply the native PID controller for Pybullet (after translating $s'$ back to the configuration space). For stability, we also set $s'$ to be not too far from $s$, by clipping the vector $\|s' - s\|_2 \leq a$. Notice that the constraint $a$ is defined in *normalized configuration space*, and in our experiments it is set to 0.03 (tuned by observing the behavior of Pybullet). The value of $a$ can be modified easily in our code.

3. The goal region $G$ is defined in three ways: (1) *EE*, *config*, and *combined*. The *EE* describes the goal as the position of the EE in $\mathbf{R}^3$, and the transition function stops the episode once this distance is small (2cm in our experiments). The *config* defines the goal as a normalized configuration, and the goal is considered reached when the normalized distance $\|g - s\|_2 \leq 0.05$. The *combined* goal representation, concatenates both representations, but defines the goal reaching predicate exactly the same as the *EE* goal representation, i.e. distance in EE positions. The goal representation, and the radius around the goal are both configurable in our framework.

4. The starting and goal configurations are sampled randomly at the start of each episode from feasible configurations. Feasible configurations are not in collision, and describe joint values that the simulator can hold for extended periods of time (to avoid setting goals that the simulator cannot reach). This selection process induces the distribution $\rho_0$.

5. The transition function $P$, is deterministic and defined by the goal representation and tolerance (described above), as well as a configurable flag `stop_on_collision`, that if set to `True` stops the episode if a collision occurred. Some NMP previous works used `stop_on_collision=False` (Strudel et al., 2021), while others used `stop_on_collision=True` (Jurgenson & Tamar, 2019). In our initial experiments, we set it to `False`, as we observed that data collection becomes even harder when setting this value to `True`.

6. The horizon $T$ is set to 400 steps, by observing trajectories from SBMP in our tasks. If the episode reaches an earlier termination (see $P$ above), a sink state $s_\perp$ can be introduced such that irrespective of the action the next state remains $s_\perp$ (and the cost $C(s_\perp, \cdot, \cdot) = 0$ for every goal and action).

A final note, we opted for minimal code modifications for the learning algorithms as possible. Thus we tried using default parameters in Huang et al. (2022) including setting $\gamma$ in SAC and TD3 essentially making the finite horizon MDP into a discounted-horizon MDP.

## C  Extended related work

**Jurgenson & Tamar (2019):**   This work introduced the DDPG-MP algorithm, a combination of the model-free DDPG algorithm with a pre-trained world model, and described the *demonstration-injection* feature. Jurgenson & Tamar (2019) shares many similarities with our work (investigate both RL and BC methods, and exploration solution), but perhaps the main concern in DDPG-MP is its oversimplification; the robotic arm is only allowed to move on the XZ *plane*, essentially making the problem 2D. This allowed the easy procedural generation of narrow passages, at the cost of making the problem less relatable to real scenarios. In our work we try to make the simulated scenarios as realistic as possible creating factory-inspired scenes. Another crucial technical aspect is the technical stack - which in Jurgenson & Tamar (2019) was based on OpenRAVE (Diankov, 2010), a more complicated simulation environment than Pybullet (Coumans & Bai, 2016–2022), which introduces another barrier of entry to the NMP practitioner.

**Strudel et al. (2021):**   The core algorithm in this work is a combination of the model free RL algorithm SAC (Haarnoja et al., 2018) with HER (Andrychowicz et al., 2017) and a PointNet (Qi et al., 2017a) encoder. This combination (SAC, HER, and PointNet) is also explored in this work (although we use PointNet++ (Qi et al., 2017b), an improvement of PointNet from the same authors). However, the solution presented in Strudel et al. (2021) only considers fixed joint robots, i.e. can be described by a single frame of reference, and their solution rely on normalizing the point clouds into that single reference frame. It is not clear how to handle robots (such as robotic arms) that have moving joints and thus several frames of reference with the proposed solution. Finally, despite the impressive results, it appears that in many of the generated environments a trajectory that bypasses the obstacles from the side can easily connect many starts to many goals. Indeed in our work, for the scenario presented here, we see that the combination of SAC, HER and PointNet++ is not as successful. This demonstrates the difficulty of assessing NMP results, and is the reason we argue that NMP environments should be grounded by relatable realistic tasks that include some simple evaluation metric such as our `go-to-goal` policy.

**Yamada et al. (2021); Liu et al. (2022):**   In these works a SAC policy is trained with the help of a SBMP, and although the integration of the RL algorithm with the SBMP is different from our methods, both solutions aim to solve the sparse rewards problem in NMP. Our approach of injecting full pre-computed demonstrations tries to remove the expensive online SBMP planning time from the training loop of the RL agent. One major difference from our work, is the focus on manipulation scenarios in Yamada et al. (2021); Liu et al. (2022). Our motivation in the RoboArm-NMPenvironment is that first NMP should be evaluated and solved on MP tasks that include challenging obstacle configurations, and only then be incorporated into a task and motion planning setup that might include various manipulation aspects.

## D  Collection of demonstrations

For every scenario we collected 10K demonstrations. To collect a new demonstration we use the RRT-connect (Kuffner & LaValle, 2000) algorithm to find motion plans. Once a plan is found, we need to verify it's feasibility with the Pybullet environment: we treat the plan as sub-goals and we follow them until the goal is reached. The collection process can fail / timeout in both parts of this computation, and we make at most 3 attempts per $(s, g)$ query before rejecting and moving on to a new query.

## E  Training and verifying the encoders

As mentioned in the main text, in our baselines we use VAE (Kingma & Welling, 2013) and VQ-VAE (Van Den Oord et al., 2017) to encode images, and PointNet++ (Qi et al., 2017b) to encode point clouds. The input to all models comes from four sensors placed around the robot in order to mitigate partial-observability issues as much as possible. All encoders were trained data (images and point clouds) gathered from 200K environments. Images are RGB images of $224 \times 224$, and each point cloud sensor samples 10K points, and returns those found in collision with objects. For completeness, we now describe how an observation from four sensors is encoded to a latent representation:

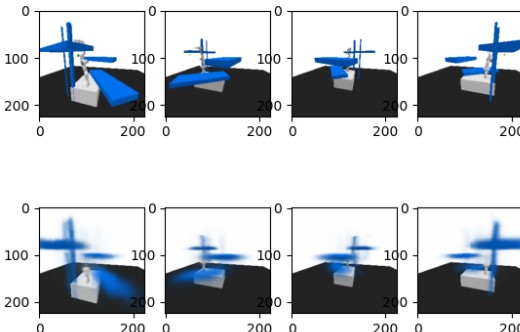

Figure 7: VAE reconstruction: top row: four ground truth images from a *random_boxes_hard* query, bottom row: reconstruction by our VAE encoder.

- VAE encoder - This encoder is trained with the VAE loss function (Kingma & Welling, 2013). We match every sensor around the robot with a VAE encoder and decoder networks, denoted by $\forall i \in \{1 \ldots 4\} : e^i_{VAE}, d^i_{VAE}$ respectively. To encode an image $x^i$ we first encode the image $z^i = e^i_{VAE}(x^i)$ to a latent vector of size 16. Then, we concatenate $z_E = [z^1, \ldots, z^4]$ (a latent vector of size 64, and the encoding of our environment $E$) and feed $z_E$ to the decoder.

- VQ-VAE (Van Den Oord et al., 2017) - similar to VAE, we have four pairs of encoder-decoders but with the VQ-VAE architecture and loss function (Van Den Oord et al., 2017). Also similar to the VAE encoder, we produce $z^i$ and $z_E$, as independent and concatenated outputs of the encoders. However, modifying the VQ-VAE decoder to infer from the combined $z_E$ was problematic as many architectural choices were required and this was out of scope of our experiments, so here opted to use $z^i$ as the input to $d^i$ directly.

- PointNet++ (Qi et al., 2017b) - To train the PonitNet++ encoder, we defined a point cloud classification task, where model tries to predict three classes of point labels: (1) points on the robot, (2) points on the table and ground plane (objects shared in every $E$), and (3) points on other obstacles (that vary with every $E$ between episodes). From the same reasons as VQ-VAE, each pair of encoder-decoder learns a representation $z^i$ independently, and only for the policy down the line we concatenate all to a single $z_E$ vector.

For all encoding schemes, we verify the model by both observing convergence of the loss plots and by visualizing the results of the prediction tasks: in VAE and VQ-VAE we reconstruct the images (Figures 7 and 9), and in PointNet++ we classify a point cloud according to the three categories above (Figure 8).

Finally, all of our latent spaces are 64-dimensional vectors, because during development when we tried training with a larger context vector, both RL and BC algorithms were unstable. Although we can see missed predictions, the visualizations show that there seems to be enough signal to understand important concepts about the 3D scene even when the latent vector is only of size 64.

## F  Prediction times investigation

We compare the inference time of our models (Table 4). We measure the average *encoding* time and the average *action prediction* time. For all our experiments, where the scene does not change within an episode, encoding is only required once, at the start of each episode, when the encoder is applied to the first sensor observation, while action prediction time denotes the forward pass of the policy network, and is required for every step of the episode. The sample was taken over 1K episodes, using our fully trained models in

| | avg. encoding time | avg. action prediction time | avg. total prediction time |
|---|---|---|---|
| VAE | 0.0045 | 0.0005 | 0.1587 |
| VQ-VAE | 0.0083 | 0.0005 | 0.1570 |
| PointNet++ | 0.0777 | 0.0005 | 0.2334 |

Table 4: Prediction times (sec) for fully trained *obstacle-generalization task* policies. We measure the average encoding time and the average action prediction time.

the *random boxes hard* task. From Table 4, we can see that deep learning solutions can start acting almost instantly, a desirable property in many domains, and that total prediction time *for the entire trajectory* is less than 0.25 seconds on average[10]. We remark that a trained NMP policy can be directly used in scenes with changing obstacles, but this requires performing the encoding step multiple times. Time measurements suggest that this approach can achieve real-time obstacle avoidance at around 100Hz.

## G   Software package description

*RoboArm-NMP* is built on top of Pybullet (Coumans & Bai, 2016–2022), and extends `Panda-gym` Gallouédec et al. (2021) designed for simple robotic manipulation tasks. The software is composed of the following directories `envs`, `demo_generation`, and learning related directories. **Code will be made available upon acceptance.**

The `envs` directory contains the logic for the environment (defined as an OpenAI gym environment (Brockman et al., 2016)), with the added api for the method `reset_specific` that unlike the `reset` method, allows the use to reset the agent to specific location. This method is used both for testing throughout the code, and for SBMP implementations for edge traversal. Moreover, various aspects of the environment can be controlled via the constructor of the object, such as the reward signal, termination condition, goal definition etc'. In this directory we also define Pybullet sensors, and wrappers to the basic OpenAI gym environment that uses those sensors. Finally, the data files and demonstration files are kept in `envs/data` and `envs/demos` respectively.

The `demo_generation` directory contains the data-collection logic for SBMP. It handles demonstration generation, verification, and correction.

The learning-related code is found in `bc_script` (for BC learning), `clean_rl_scripts` (for RL learning, extends Huang et al. (2022)), and `scene_embedding_train` (for learning visual encoders).

## H   Extending RoboArm-NMP

We included a document in our repository, `RoboArmNMP_code_usage.pdf`, for extending the benchmark with new scenes (and their demonstrations), sensors, encoders for sensors, and learning algorithms.

## I   Parameter sweeps

For TD3 we used the default parameters from Huang et al. (2022), and created a sweep for policy and Q-network learning rates. Both values were between 0.005 and 0.000001 and executed on the *double walls wide gap* task. For SAC we searched over the learning rates like we did for TD3, and we also tried to enable / disable the entropy auto-tuning, and the value of $\alpha$ between 10 to 0.00001.

For BC, we searched over the learning rate (similar range to TD3 and SAC), and we also tried to increase the capacity of the policy by learning a Gaussian mixture-model instead of a uni-modal Gaussian. We found that a uni-modal prediction works best.

---

[10]For comparison in Schulman et al. (2014) in a similar setting for a 7DoF robot SBMP took over 0.6 seconds to compute the trajectory on average.

| | no obstacles | wall | double walls wide gap | double walls | boxes |
|---|---|---|---|---|---|
| Go-to-goal | 0.9778 | 0.7646 | 0.6476 | 0.4521 | 0.806 |
| BC | $0.985 \pm 0.004$ | $0.752 \pm 0.004$ | $0.638 \pm 0.008$ | $0.500 \pm 0.009$ | $0.769 \pm 0.012$ |
| SAC (no demos) $p = 0$ | $0.994 \pm 0.002$ | $0.948 \pm 0.003$ | $0.866 \pm 0.006$ | $0.669 \pm 0.026$ | $0.849 \pm 0.003$ |
| SAC-MP $p = 0.2$ | $0.992 \pm 0.001$ | $0.942 \pm 0.005$ | $0.879 \pm 0.004$ | $0.730 \pm 0.008$ | $0.859 \pm 0.005$ |
| SAC-MP $p = 0.5$ | $0.993 \pm 0.001$ | $0.947 \pm 0.003$ | $0.880 \pm 0.006$ | $0.769 \pm 0.020$ | $0.861 \pm 0.004$ |
| SAC-MP $p = 1.0$ | $0.993 \pm 0.001$ | $0.936 \pm 0.003$ | $0.878 \pm 0.006$ | $0.798 \pm 0.011$ | $0.857 \pm 0.001$ |
| TD3 (no demos) $p = 0$ | $0.992 \pm 0.002$ | $0.924 \pm 0.010$ | $0.767 \pm 0.025$ | $0.483 \pm 0.014$ | $0.816 \pm 0.012$ |
| TD3-MP $p = 0.2$ | $0.993 \pm 0.001$ | $0.920 \pm 0.011$ | $0.752 \pm 0.014$ | $0.547 \pm 0.013$ | $0.845 \pm 0.016$ |
| TD3-MP $p = 0.5$ | $0.991 \pm 0.002$ | $0.916 \pm 0.013$ | $0.775 \pm 0.038$ | $0.565 \pm 0.044$ | $0.793 \pm 0.029$ |
| TD3-MP $p = 1.0$ | $0.988 \pm 0.001$ | $0.920 \pm 0.003$ | $0.815 \pm 0.011$ | $0.625 \pm 0.047$ | $0.822 \pm 0.014$ |

Table 5: Goal-generalization tasks with goal type *configuration*

## J  Additional results for the `obstacle generalization tasks`

In this section we report the results of additional experiments that were part of our investigation.

**The `goal generalization tasks`:** We provide full results for the goal various goal encoding schemes in Tables 5, 6, and 7.

**Additional `obstacle generalization` results:** In Table 7 we see the full results of our `obstacle generalization` tasks, including the less effective PointNet++ and VAE encoders.

Moreover, we investigated transfer to `obstacle-generalization tasks` with less obstacles. We find that the policies trained in *random boxes hard*, transfer well to both *random boxes medium* and *random boxes easy*. In *random boxes medium*, that is more similar to *random boxes hard*, both SAC and SAC-MP ($p = 0.5$) were able to reach higher success rates than the `go-to-goal` policy. This result is meaningful because it hints that if we train our agents in complex enough environments, they will be able to utilize the same policy in simpler situations as well.

| | no obstacles | wall | double walls wide gap | double walls | boxes |
|---|---|---|---|---|---|
| Go-to-goal | 0.9794 | 0.7647 | 0.648 | 0.4528 | 0.806 |
| BC | $0.073 \pm 0.036$ | $0.043 \pm 0.012$ | $0.027 \pm 0.009$ | $0.023 \pm 0.005$ | $0.033 \pm 0.017$ |
| SAC (no demos) $p = 0$ | $0.969 \pm 0.010$ | $0.941 \pm 0.012$ | $0.853 \pm 0.015$ | $0.730 \pm 0.031$ | $0.851 \pm 0.020$ |
| SAC-MP $p = 0.2$ | $0.977 \pm 0.003$ | $0.963 \pm 0.005$ | $0.925 \pm 0.005$ | $0.883 \pm 0.016$ | $0.905 \pm 0.009$ |
| SAC-MP $p = 0.5$ | $0.975 \pm 0.006$ | $0.971 \pm 0.003$ | $0.941 \pm 0.002$ | $0.896 \pm 0.010$ | $0.912 \pm 0.005$ |
| SAC-MP $p = 1.0$ | $0.979 \pm 0.001$ | $0.971 \pm 0.003$ | $0.944 \pm 0.008$ | $0.916 \pm 0.004$ | $0.922 \pm 0.003$ |
| TD3 (no demos) $p = 0$ | $0.976 \pm 0.003$ | $0.942 \pm 0.005$ | $0.896 \pm 0.012$ | $0.598 \pm 0.029$ | $0.661 \pm 0.381$ |
| TD3-MP $p = 0.2$ | $0.973 \pm 0.003$ | $0.961 \pm 0.006$ | $0.922 \pm 0.013$ | $0.824 \pm 0.026$ | $0.906 \pm 0.006$ |
| TD3-MP $p = 0.5$ | $0.975 \pm 0.004$ | $0.964 \pm 0.003$ | $0.934 \pm 0.005$ | $0.643 \pm 0.371$ | $0.902 \pm 0.011$ |
| TD3-MP $p = 1.0$ | $0.968 \pm 0.004$ | $0.959 \pm 0.002$ | $0.918 \pm 0.010$ | $0.848 \pm 0.027$ | $0.903 \pm 0.011$ |

Table 6: Goal-generalization tasks with goal type *end-effector*

| | no obstacles | wall | double walls wide gap | double walls | boxes |
|---|---|---|---|---|---|
| Go-to-goal | 0.9794 | 0.7647 | 0.648 | 0.4528 | 0.806 |
| BC | $0.986 \pm 0.002$ | $0.831 \pm 0.023$ | $0.656 \pm 0.079$ | $0.463 \pm 0.107$ | $0.771 \pm 0.054$ |
| SAC (no demos) $p = 0$ | $0.990 \pm 0.002$ | $0.933 \pm 0.009$ | $0.858 \pm 0.008$ | $0.677 \pm 0.015$ | $0.827 \pm 0.004$ |
| SAC-MP $p = 0.2$ | $0.991 \pm 0.002$ | $0.939 \pm 0.005$ | $0.868 \pm 0.016$ | $0.748 \pm 0.025$ | $0.842 \pm 0.013$ |
| SAC-MP $p = 0.5$ | $0.989 \pm 0.002$ | $0.938 \pm 0.009$ | $0.876 \pm 0.009$ | $0.763 \pm 0.001$ | $0.851 \pm 0.007$ |
| SAC-MP $p = 1.0$ | $0.989 \pm 0.002$ | $0.943 \pm 0.007$ | $0.886 \pm 0.011$ | $0.796 \pm 0.023$ | $0.862 \pm 0.006$ |
| TD3 (no demos) $p = 0$ | $0.977 \pm 0.007$ | $0.823 \pm 0.042$ | $0.733 \pm 0.055$ | $0.429 \pm 0.014$ | $0.726 \pm 0.047$ |
| TD3-MP $p = 0.2$ | $0.976 \pm 0.005$ | $0.902 \pm 0.018$ | $0.745 \pm 0.030$ | $0.515 \pm 0.017$ | $0.808 \pm 0.019$ |
| TD3-MP $p = 0.5$ | $0.980 \pm 0.002$ | $0.906 \pm 0.004$ | $0.730 \pm 0.035$ | $0.565 \pm 0.036$ | $0.815 \pm 0.032$ |
| TD3-MP $p = 1.0$ | $0.961 \pm 0.012$ | $0.889 \pm 0.020$ | $0.633 \pm 0.079$ | $0.523 \pm 0.032$ | $0.738 \pm 0.087$ |

Table 7: Goal-generalization tasks with goal type *combined*

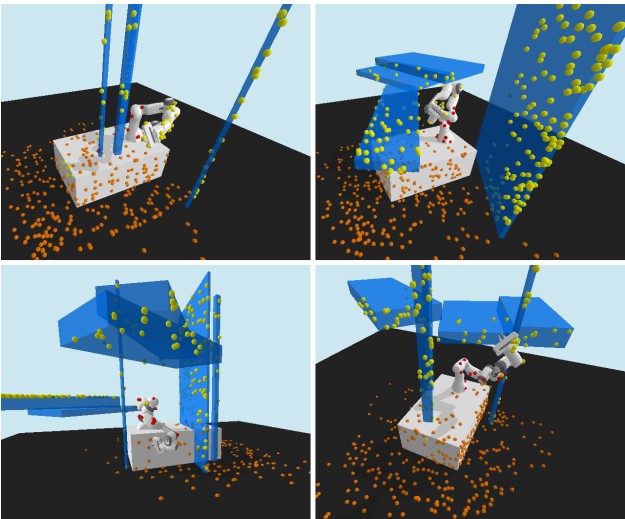

Figure 8: PointNet++ classification: spheres color denotes the class *predicted* by PointNet++. The colors red, orange, and yellow, correspond to predictions on the robot, static obstacles, and varying obstacles respectively.

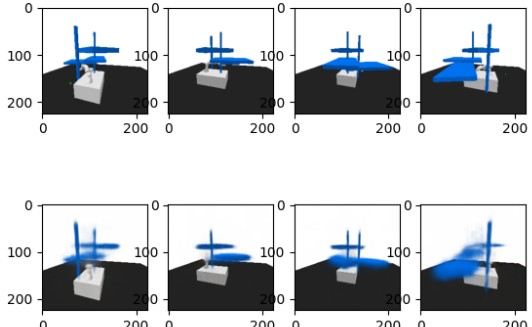

Figure 9: VQ-VAE reconstruction: top row: four ground truth images from a *random_boxes_hard* query, bottom row: reconstruction by our VQ-VAE encoder.

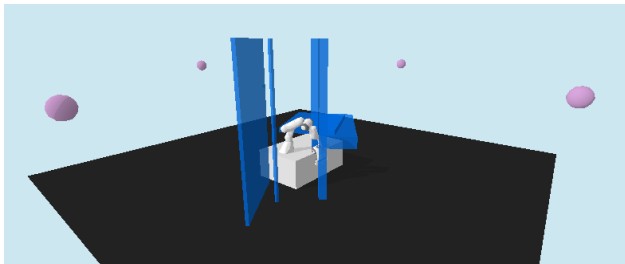

Figure 10: Visualization of sensor placements in the *obstacle-generalization tasks*. Each sensor position is marked with a purple sphere.

| | random boxes hard (trained) | random boxes medium | random boxes easy | three shelves | pole shelves | narrow shelves |
|---|---|---|---|---|---|---|
| VAE (no-demos) | $0.284 \pm 0.003$ | $0.366 \pm 0.005$ | $0.467 \pm 0.009$ | $0.771 \pm 0.025$ | $0.726 \pm 0.017$ | $0.466 \pm 0.022$ |
| VAE $p = 0.5$ | $0.234 \pm 0.024$ | $0.306 \pm 0.032$ | $0.397 \pm 0.046$ | $0.626 \pm 0.124$ | $0.613 \pm 0.074$ | $0.449 \pm 0.039$ |
| VAE $p = 1.0$ | $0.199 \pm 0.012$ | $0.262 \pm 0.017$ | $0.333 \pm 0.024$ | $0.543 \pm 0.067$ | $0.531 \pm 0.103$ | $0.353 \pm 0.058$ |
| VQ-VAE (no-demos) | $0.304 \pm 0.007$ | $0.380 \pm 0.009$ | $0.483 \pm 0.012$ | $0.792 \pm 0.009$ | $0.721 \pm 0.020$ | $0.493 \pm 0.026$ |
| VQ-VAE $p = 0.5$ | $0.302 \pm 0.008$ | $0.388 \pm 0.012$ | $0.495 \pm 0.017$ | $0.816 \pm 0.015$ | $0.741 \pm 0.024$ | $0.497 \pm 0.043$ |
| VQ-VAE $p = 1.0$ | $0.274 \pm 0.009$ | $0.347 \pm 0.011$ | $0.445 \pm 0.014$ | $0.750 \pm 0.028$ | $0.696 \pm 0.015$ | $0.405 \pm 0.022$ |
| PointNet++ (no-demos) | $0.279 \pm 0.007$ | $0.351 \pm 0.005$ | $0.444 \pm 0.009$ | $0.635 \pm 0.068$ | $0.666 \pm 0.016$ | $0.486 \pm 0.031$ |
| PointNet++ $p = 0.5$ | $0.254 \pm 0.032$ | $0.325 \pm 0.046$ | $0.415 \pm 0.056$ | $0.662 \pm 0.083$ | $0.624 \pm 0.067$ | $0.469 \pm 0.076$ |
| PointNet++ $p = 1.0$ | $0.240 \pm 0.016$ | $0.310 \pm 0.020$ | $0.390 \pm 0.031$ | $0.573 \pm 0.048$ | $0.598 \pm 0.041$ | $0.441 \pm 0.019$ |
| Go-to-goal | 0.2891 | 0.3697 | 0.4914 | 0.8316 | 0.7207 | 0.5393 |

Table 8: Policy transfer: we compare our models trained only on *random boxes hard* on (1) two easier yet similar tasks *random boxes medium*, and *random boxes easy*, and (2) on three hand-crafted test tasks *three shelves*, *pole shelves*, and *narrow shelves*.

