# OpenReview forum: "RoboArm-NMP: a Learning Environment for Neural Motion Planning"
_TMLR — Rejected by TMLR_

### Review · Reviewer_B6WT · 2024-01-08

**Summary Of Contributions:**

The paper presents a software framework, a set of benchmark tasks, and an empirical study of Neural Motion Planning (NMP) algorithms.

**Audience:**

Yes

**Broader Impact Concerns:**

There is no broader impact statement included in the paper

**Claims And Evidence:**

Yes

**Requested Changes:**

See "Weaknesses" of the review

**Strengths And Weaknesses:**

The paper studies a relevant problem: What matters for NMP algorithms and how do we measure the various factors? It fills a gap in existing literature: As detailed in the paper, existing motion planning benchmarks lack important functionality required for evaluating NMPs, such as demonstration dataset and corresponding utilities for encoding observations and facilitating RL-based training. The empirical study is comprehensive and covers important traits of NMPs such as scene encoders, goal specifications, and the use of demonstrations.

While I appreciate the thoughtful construct of the study and the effort, I believe the paper can benefit from addressing a few issues:

Scenario realism. Does the environments included in the framework reflect real-world distributions of obstacles and constraints for plan solutions? And does this realism matter? It’d be great to at least draw a few environments from real-world rooms / scenarios such as the House3D dataset or discuss why the environments included now are sufficient.

Evaluation metric: Success rate is a very coarse metric. For example, analyzing the cost of successful solutions and qualitatively comparing that of the ground truth planner, the RL algorithm, and the BC algorithm may give valuable insights about the characteristics of the models.

Evaluation of Learned Planning Space: There is no real analysis of observation encoders and the resulting planning space. What matters in the various design choices of the encoder? For example, what kind of embedding space does VQ-VAE learn and why does it allow for better performance for BC compared to VAE? Does 3D input provide more useful information for the downstream NMP?

Is there a particular reason that the work focuses on a fixed-base manipulator as opposed to a mobile manipulator? The latter seems to introduce more interesting challenges (partial observability, base obstacle avoidance, larger robot state space etc.)

Finally, it’d be great to include the planning speed for the reference ground truth motion planner. At the end of the day, the advantages of NMP boils down to speed and end-to-end representation learning.

---

> ### Author Response · Authors · 2024-02-01
> **Response**
>
> We thank you for your time and constructive review.
>
> Regarding scenario realism, our motivation is creating a useful and challenging distribution of workspaces where there are many cluttered objects. We hypothesize that successfully navigating such a world would be conducive to solving real world scenarios where obstacles are probably less cluttered. In most recent NMP works the scenes are not challenging nor realistic enough, or the task revolves around motion planning while executing a manipulation task (see the Related work section for more elaborate description of current limitations).
>
> Regarding Houses3D and similar benchmarks - these are mostly challenging for navigation, or motion planning and navigation (a robotic arm on top of a mobile base). For motion planning of a static arm, which is the focus of this work, the scenes in Houses3D do not appear to be more challenging, and are less cluttered, than our environments.
>
> Finally, regarding the question of “does realism matter?” We think that it does, but in this study we limited ourselves to simulated scenarios because, as we showed, they provide a substantial challenge for modern day NMP algorithms even without considering the simulation-to-reality gap, and they have the benefit of being copied and deployed easily compared to real-world setups.
>
> Evaluation metrics - we mostly focused on success rates, because our paper indicates that even being successful in motion planning is challenging. Other metrics (such as path length or minimal clearance) could be easily added, but we think that they could distract from the main results of identifying the low success rates of modern NMP algorithms. If the reviewer thinks it is still a crucial addition we could add those metrics.
>
> We shall include the planning speed of our planners, but we don’t agree with the statement that it boils down to speed of NMP algorithms compared to sampling based planners -- after training sampling based planners still require precise knowledge of the environment, while NMP policies do not. This is a crucial benefit of these systems.
>
>
> Regarding the embedding space, it is an interesting question, and during our evaluation we have only verified that a simple classifier predicting if a coordinate is free or not (i.e. a function F(x,y,z,E_z) where E_z is the embedding vector) is performing well enough.
> We could include these results as well as the following analysis (on frozen visual encoders):
> 1. Investigating how small perturbations in the parameters of the scene affect the embedding vectors.
> 2. Predicting from the embedding vectors quantities such as the number of objects in the scene.

---

> > ### Comment · Reviewer_B6WT · 2024-02-13
> > **Follow up to  author's responses**
> >
> > Thanks for the response.
> >
> > Scenario realism: I don't think the authors have adequately addressed my concern regarding scenario realism. My original question is simply: How can you convince the readers that the conclusions drawn in this particular benchmark environment can transfer to scenes with similar obstacle distributions in the real world? I agree that introducing challenging and cluttered environments is a valid and important feature of this benchmark, and sometimes existing works favor "easier solutions". But this still doesn't establish the critical link between [findings in this benchmark] and [findings in realistic environments].
> >
> > Evaluation metrics: It'd be ideal to include cost metrics in addition to the success rate.
> >
> > Including additional ablation studies on embedding space, including the ones suggested by the authors, will be idela.

---

> > > ### Author Response · Authors · 2024-02-21
> > > **We thank the reviewer for further clarifying their concerns**
> > >
> > > We thank the reviewer for further details regarding the review.
> > >
> > > Regarding scenario realism, i.e. the connection between
> > > [findings in this benchmark] and [findings in realistic environments], we do not agree with the reviewer on this point.
> > > It is true that [findings in this benchmark] do not necessarily translate to performance on any real world problem. In this sense, this benchmark is not “useful” as is.
> > >
> > > However, we are not sure *what is* a good distribution of real world problems that could replace our environments. If we focused on, say, “office scenes”, one could argue that solving this won’t generalize to “warehouse scenes”. There are many possible “real world” problems, and we do not know how to cover them in a convincing manner, and more importantly, how to claim that any proposed task distribution covers them “enough”. In addition, we showed that *some* previous work on so called “real world” tasks resulted in too simple environments. So we are not sure what is the concrete change that the reviewer suggests.
> > > Instead, we designed a suite of tasks where we *can control the difficulty level*, by means of the number of objects and the clearance between them - concepts that are known to correlate with the difficulty of the motion planning problem (at least for sampling based planners, which we aim to improve upon). Solving our tasks will necessarily lead to better neural motion planning algorithms, which is the goal of our benchmark.

---

### Review · Reviewer_wZ3s · 2024-01-18

**Summary Of Contributions:**

This paper develops RoboARM-NMP, a neural motion planning (NMP) benchmark (set of virtual tasks for a 7-DoF robot manipulator (static base) to perform). This tool is developed for the community to meet a need: the paper claims it is difficult to compare various baselines today due to a lack of an appropriate benchmarking environment. The paper then compares such baselines in the benchmarking environment and makes observations regarding the efficacy of demonstration data and hindsight experience replay.

**Audience:**

Yes

**Claims And Evidence:**

No

**Requested Changes:**

Kindly respond to the list of strengths and weaknesses above and the questions, etc. below.

Questions:
-"Thus, at present, it is difficult to assess the fundamental capabilities of NMP, namely generalization to unseen goals or obstacle configurations, performance on ‘difficult’ instances such as narrow passages, and how to tease out the essential algorithmic ideas that make NMP work in general." Are the authors trying to say that these problems (after "namely") are the ones that are specifically difficult to address because of a lack of a benchmarking environment or are these attributes that are important to address in general and having a benchmarking environment would be helpful to do so? Why are these the important/only ideas considered in this paper?

- Similarly, section 5 identifies three key questions. What questions are left off this list or out of scope of the paper?

-The 7-DoF robot appears to be fixed to a static base rather than on a mobile platform. What is the inspiration for this limitation?

Comments:
-Figure 1 could be improved by showing an trajectory of sorts rather than still frames from different environments (or one could show multiple trajectories across different environments). It is hard to know what is happening in each environment, e.g. the start and goal locations, kinematics, etc.

Writing:
-"obstacles, but" should not have a comma
-"comparing the various different" should be "comparing different" unless the authors are meaning a specific set of NLP algorithms is difficult rather than comparing them in general.
-No comma here: "configurations), with" unless the next phrase after with is not a restriction of the first phrase but instead a completely separate item. If the latter, then the phrase needs to be re-written for grammatical consistency.
-No comma before and here: "positions), and,"
-Comma needed after "To fill these gaps"
-Spell out all acronyms for the first use (e.g., "DoF")
-No comma here: "investigating NMP, by choosing." I recommend the authors review the rules for punctuation/grammar for restrictive clauses versus non-restrictive clauses.
-No comma here: "in Python, and builds." This reviewer will not comment further on comma splices, etc. and kindly asks the authors to address level-2 editing needs for this paper.

**Strengths And Weaknesses:**

Strengths:
-This paper addresses an important problem in robotics by developing a benchmarking environment for neural motion planning.
-This paper conducts an appropriate set of empirical evaluations of various NMP techniques in this benchmarking environment.
-The paper articulates key metrics for NMP algorithms.
-Section 2 covers a nice breadth of topics in a concise manner for the preliminaries needed to understand the paper.

Weaknesses:
-The paper appears to claim on page two that a contribution of this paper is that this work considers both demonstrations and hindsight and shows the combination is essential. However, this paper does not appear to be the first to use this combination in general (see below). Perhaps this claim can be valid if caveats are added.

Ding, Y., Florensa, C., Abbeel, P. and Phielipp, M., 2019. Goal-conditioned imitation learning. Advances in neural information processing systems, 32.

Liu, N., Lu, T., Cai, Y., Li, B. and Wang, S., 2019. Hindsight generative adversarial imitation learning. arXiv preprint arXiv:1903.07854.

Yu, X., Bai, C., Wang, C., Yu, D., Chen, C.P. and Wang, Z., 2023. Self-Supervised Imitation for Offline Reinforcement Learning With Hindsight Relabeling. IEEE Transactions on Systems, Man, and Cybernetics: Systems.

-The paper's literature review (Section 3) some of the appropriate works (though notable exceptions above); however, most of these works are not actually discussed in sufficient detail to understand general NMP approaches, what their limitations that necessitates further work in the area and good benchmarking tools, and what existing benchmarks -- even if had hoc -- are available and why those are inadequate. In particular, more discussion and justification w.r.t. references, such as Chamzas et al. (2021), would be helpful. Section C in the appendix is nice but not sufficient to address these concerns, and Section C should be integrated into the main paper. This reviewer is concerned that the contribution and novelty here may be limited; further, the reviewer is not convinced that NMP needs specific benchmarks different from symbolic MP.

-The paper appears to claim that out-of-distribution (OOD) is synonymous with "markedly different," which is not a technical specification. It would be helpful if the paper clearly defined with metrics what OOD means in the context of NMP.

-For a paper developing a benchmark, the description of the environment (Section 4) seems relatively superficial. More detail to understand the expertise and considerations that went into designing this benchmark would be important to motivate why researchers in NMP should adopt this dataset. For example, how do readers know that this benchmarking set captures important aspects of real-world problems? If a researcher does well on this benchmark, what problems in what industries in the real-world can now be solved?

-Figure 3 is hard to read with tiny text and unclear labels. It is difficult to assess overall pros/cons of baselines across these plots (i.e., trends). Also, the results could be improved in presentation if graphics could accompany these bar graphs that show a representative testing domain for each subfigure.

- The paper makes numerous claims and adds commentary about the results without performing an analysis of variance (ANOVA). It would be helpful, particularly with so many different tests performed, to follow best practices in statistical analysis and controlling for random chance and follow-through on a thorough statistical analysis. In particular, this analysis could support claims about demonstration data and hindsight experience replay. This analysis could also help directly "test" the research questions posed at the beginning of Section 5.

-This reviewer expected more content about ease of use, usability (user-centered design), and other validation (e.g., limited deployments of the benchmark for potential users) to motivate why other researchers should be eager to adopt this benchmarking environment.

- Section 7 can be improved by adding more content that clearly draws upon the empirical results and calls out specific design guidelines and needs and how this benchmarking tool will enable researchers to address those needs.

- It would be helpful to fold key results from the appendix into the paper and the statistical analysis so that a clear consensus and set of take-aways can be developed rather than leaving the reader to try to discern approximate trends.

---

> ### Author Response · Authors · 2024-02-01
> **Rebuttal response**
>
> We thank the reviewer for taking time to review our paper, and for their thoughtful suggestions and insights. We are happy the reviewer found that “This paper addresses an important problem in robotics by developing a benchmarking environment for neural motion planning”, which was our main goal in this work. We next respond to the concerns raised by the reviewer:
>
> See main response regarding the applicability and simulation to real gap.
>
> Regarding the hindsight and demonstration novelty claim; although we investigated hindsight methods and demonstrations, these are only examples that illustrate why our learning environment is required (by showing that even with well-established methods NMP algorithms still have very low success rates such as those you suggested in [1-3]). We want to emphasize that we did not claim the results of this investigation as novel findings, but we will make our contribution clearer.
>
> Next we respond to the related works comments: First, regarding section C, we will integrate it into the main paper, but it would be helpful to know why the reviewer thinks the section is only “nice but not sufficient”?
>
> Regarding symbolic MP, as we mentioned in the paper (page 1 and 11), there are two reasons for preferring our environments, the first is the python integration which is an essential component when developing ML and RL solutions, and the second is the dataset we collected that can be used to train agents that facilitates the entire ML approach.
>
> *"The paper appears to claim that out-of-distribution (OOD) is synonymous with "markedly different," which is not a technical specification. It would be helpful if the paper clearly defined with metrics what OOD means in the context of NMP.”*
>
> Not quite so, in our context OOD means that the train and test distributions of obstacle parameters are not the same. E.g., the number of obstacles is different, but not the obstacles themselves. Our obstacle primitives, which are walls, poles (columns) and shelves (planes) are frequently encountered in other environments and in real scenarios. We imported some scenarios from other environments, and we used these primitives to design realistic test-cases.
>
> Regarding ease of use, we have included a developer guide. Please see RoboArmNMP_code_usage.pdf (the file is in the git repo because of it being too technical for the paper).
>
> Regarding the Figure 3 and folding key insights into the main paper - we thank the reviewer for the suggestion we will do so.
>
> [1] Ding, Y., Florensa, C., Abbeel, P. and Phielipp, M., 2019. Goal-conditioned imitation learning. Advances in neural information processing systems, 32.
>
> [2] Liu, N., Lu, T., Cai, Y., Li, B. and Wang, S., 2019. Hindsight generative adversarial imitation learning. arXiv preprint arXiv:1903.07854.
>
> [3] Yu, X., Bai, C., Wang, C., Yu, D., Chen, C.P. and Wang, Z., 2023. Self-Supervised Imitation for Offline Reinforcement Learning With Hindsight Relabeling. IEEE Transactions on Systems, Man, and Cybernetics: Systems.

---

> > ### Comment · Reviewer_wZ3s · 2024-02-12
> > **Response**
> >
> > Regarding the "we will make our contribution clearer." for the discussion of novelty, this reviewer looks forward to seeing this change in a revision.
> >
> > Section C of the appendix discusses some related works in a helpful level of detail to contextualize this paper submission, which is why the reviewer called it "nice." The reviewer said it was "not sufficient" because it only covered a small minority of the related work, leaving the reviewer with questions about exactly why this work is novel and significant -- the details matter, and the lack of details in Section 3 were not made up for just with Section C; more details -- a more thorough and detailed literature review -- is needed.
> >
> > The authors respond by saying that this reviewer's point about the discussion of OOD was "not quite so." Yet, it is unclear that the authors actually addressed the concern. The paper literally says, "OOD tasks: We defined three fixed environments...these scenarios are markedly different from scenarios sampled as described above, and we do not allow training on them." This paragraph does not offer a clear, technical definition of OOD. No such definition appears to exist anywhere in the paper. This is a key weakness of the paper.
> >
> > Further, the author response says, "the number of obstacles is different, but not the obstacles themselves" is one way in which the tasks are OOD. Yet, it is difficult to understand how this is truly OOD. If one has a CV image segmentation task, then does having a different number of objects to segment in a test image (e.g., an image with 5 humans) than any seen in training (e.g., 1, 2, 3, 4, and 6 humans) mean that the testing scenario is an OOD task? A clear, technical definition that is supported by prior work would be important to justify that interpolation is really OOD performance.

---

> > > ### Author Response · Authors · 2024-02-23
> > > **OOD clarification**
> > >
> > > We thank the reviewer for further clarifications, and we would like to elaborate on the topic of OOD:
> > >
> > > There is some confusion here, and the reviewer’s image segmentation example is not completely fitting.
> > > OOD in machine learning is well defined in the following sense: if there is a distribution over training tasks $P(x)$, then $x’$ is OOD if $P(x’)=0$. In this case, if during training $x$ can be 1,2,3,4, or 6, then $x’=5$ is OOD just as $x’=7$ or $x’=1000$.
> > > On the contrary, it is interpolation and extrapolation in high dimensions that are not well defined concepts, not our OOD definition.
> > >
> > > Regarding the reviewer’s example - the fact that 5 seems less OOD than 1000 relates to the inductive bias in the NN architecture. In image segmentation, for example, a fully convolutional neural network architecture should, by construction, work for 5 objects if trained on 1,2,3,4,6. But in general, this is not necessarily the case. For example, in regression, fitting a linear line to a function that is linear everywhere except for a ‘bump’ in the middle, and assuming we don’t have data in the ‘bump’, will have perfect extrapolation outside the bump but bad interpolation in the bump.
> > > We agree, however, that there are many forms of OOD tasks, and our OOD experiments, even though well defined, are not indicative of other OOD tasks. We will clarify this and tone down our claims.

---

### Review · Reviewer_t8RD · 2024-01-19

**Summary Of Contributions:**

This paper presents a new benchmark, RoboArm-NMP, for investigating the differences in performance between different neural motion planning (NMP) algorithms. The paper investigates several architectural choices that could impact NMP performance, including hindsight experience replay (HER), demo-injection, goal representation, and behavioral cloning vs. reinforcement learning. Each choice is investigated in the context of the benchmark, which involves “easy”, “medium” and “hard” tasks, where the major difference in each is the number of random obstacles in the scene of different shapes. The goal is always for a 7 DOF arm to reach a goal location without colliding with any of the boxes. The paper finds that all NMP algorithms struggle to generalize across obstacle configurations, but they can generalize across goal configurations. The paper also notes several other smaller findings, such as how to choose a goal representation for behavioral cloning vs. RL, and the advantages of HER / demo injection.

**Audience:**

Yes

**Broader Impact Concerns:**

No concerns.

**Claims And Evidence:**

Yes

**Requested Changes:**

# Critical for securing my recommendation

## Visual encoder experiment changes
One of the main claims of the paper is that NMP methods cannot generalize to new obstacle configurations. However, there could be two reasons for this: either the visual encoders are not particularly good, or the policies overfit. I think the paper needs to have a better explanation for which of these is going on. In particular, while the appendix shows reconstructions from the VAE, it appears that this is only being investigated on the training data. I would like to see the reconstruction quality for the actual testing tasks that the policy is being evaluated on, in order to understand if the issue is about the visual representation, or about the policy.

## Clarification of experimental procedure
“We repeat each experiment four times and report means and standard deviations”

– Does this mean that the models were each trained four times? Or one model is trained, and then evaluated four times for each experiment? If the latter, I believe more training runs are likely needed to interpret the results of this paper. If the former, how is this done for the case of the visual encoders, where the visual encoder is trained separately? Are there 4x4 experiments (4 for each visual encoder x 4 for each policy training)?

## Reducing the claims about statistically insignificant results
The paper occasionally over-interprets results that are not statistically significant. For example:

“While demo-injection negatively affected performance in random boxes hard…”

This isn’t a statistically reliable effect, the difference in performance is miniscule relative to the standard deviation. It is important to scope the claims appropriately. Similarly, please add error bars to Figure 3 so that differences between the methods can be interpreted more clearly.

## Results for RL on end-effector positions rather than combined goal representation
In the first part of the results section, the authors point out that the best goal representation for RL is to use end-effector positions. However, for the obstacle generalization task, the “combined” representation is used instead: “To limit the number of parameters to investigate, we make some use of the conclusions from the goal-generalization experiments, and choose the combined goal representation, HER, and SAC for our RL experiments.”

– This seems like a very odd choice given the previous claim that EE worked best as the goal representation from the previous section. Why would the best representation not be used? Can the experiments be repeated with the EE representation?

# Strengthening the work
## Obstacle shape vs. configuration generalization
The paper examines goal generalization and obstacle generalization, but obstacle generalization conflates obstacle shape with obstacle position. It would be interesting to see experiments that examine obstacle generalization separately for new configurations of previously seen obstacles vs. similar configurations of new obstacle shapes.

## Figure 1 clarification
It is a bit difficult to see what is going on in Figure 1. Obstacles appear slightly transparent? I am not convinced that this is striking the right balance between showcasing clarity and diversity of the RoboArm-NMP benchmark. I would recommend either picking one scenario and showing it from multiple views, or adding a greater diversity of tasks to show the generality of the benchmark. It is also not clear from this figure what the goal is in the task. Is it to reach the green dot?

## Notation clarification
* Behavioral cloning section
  * What is “l”? Should it not be that a state trajectory has T timesteps for T actions? If not, might be worth having an explicit definition of l, and say l < T.
* Confusion between state and observation
  * Is state only the robot’s state? Or also the configuration of the environment? Separating these out for how different aspects are represented would be useful for a reader

## Obstacle state representation
It would be nice to know whether the issue with obstacle generalization is because the policy does not generalize, or because the visual encoders are not particularly good for the new obstacle configurations. One way to do this is to have the validation provided in the “must-have” section above, but an alternative would be to compare performance of these models to ones that have object state information. Concretely, a graph network or transformer could be used for this in order to handle different numbers of obstacles and still get a single latent vector.

## Random editing points
* End of page 4 citations are mis-formatted
* “the goal configuration of the robot”
What does this actually mean? Is it the full skeleton? Joint configuration at the goal?
* “ (the success rate of SAC is greater than SAC-MP with p = 0.5, which in turn is greater than SAC-MP with p = 0.5 for every visual encoder)” I assume this is a typo? Do you mean p=1.0?
  * Table 3 suggests differences are very small, and not statistically significant
* Footnote 3 does not make sense: “Obstacles do not change shape and position between episodes, the change occurs in between episodes.” This is directly contradictory
* table 2 is mentioned later in the text relative to table 3, so these should probably be switched in ordering

**Strengths And Weaknesses:**

Strengths
* Provided code appears to be well-documented and easy-to-use, so practitioners will hopefully be able to use it (although training data was not provided, so I could not run this myself to check that everything works)
* Clearly written paper with only minor issues that can be easily addressed. The paper flows well from introduction through to conclusion.
* Experiments, for the most part, are carefully and thoroughly done. A reasonable number of different design choices are studied, and some of the findings contradict existing knowledge in the field, suggesting that the community will find this paper to be valuable.

Weaknesses
* Clarity could be improved in sections, especially in how the experiments were conducted
* Some of the experiments, especially related to visual encoders, are not well controlled enough for the claims being made about them
* Some of the results are over-interpreted given that the statistical effects are weak

---

> ### Author Response · Authors · 2024-02-01
> **Response**
>
> We thank the reviewer for the thoughtful insights and comments, and for acknowledging that our “code appears to be well-documented and easy-to-use”, and that “some of the findings contradict existing knowledge in the field, suggesting that the community will find this paper to be valuable.” Regarding the issues that were raised:
>
> Visual encoders - indeed, more investigations into the learned visual encoding could be provided. See our proposed investigations in the general comment. Regarding the training set of visual scenes claim - the results presented in the paper are on scenes not seen during training, we will emphasize this better in the next version.
>
> Regarding the remaining comments: experiments were repeated (trained from scratch) over four different seeds. Moreover, we will tone down the claims about statistically insignificant results as the reviewer suggested. Finally, regarding the goal representation, due to the high cost associated with conducting a full suite of experiments we wanted a representation that works well for both RL and BC. We will clarify these points better and the improvements suggested by the reviewer.

---

> > ### Comment · Reviewer_t8RD · 2024-02-13
> > **Follow up to authors**
> >
> > I look forward to seeing the revisions as suggested. Related to the random seeds for training from scratch, can the authors please clarify then a response to this point from the initial review: "How is this done for the case of the visual encoders, where the visual encoder is trained separately? Are there 4x4 experiments (4 for each visual encoder x 4 for each policy training)?"?
> >
> > Otherwise, I look forward to seeing the additional visual encoding experiments, and the revised text.

---

> > > ### Author Response · Authors · 2024-02-17
> > > **Further clarification**
> > >
> > > Indeed as you mentioned, there are 4 for each visual encoder x 4 for each policy training experiments. But we will clarify that in the paper.
> > >
> > > We will soon upload a revised pdf with changes from all the reviewers.

---

### Author Response · Authors · 2024-02-01
**Rebuttal response**

First, we’d like to thank the reviewers for their time and thoughtful suggestions for improvements to our paper. We address some recurring topics here, and dive further into the reviewers’ comments in the individual responses.


1. Regarding *scenario realism*, our motivation is creating a useful and challenging distribution of workspaces with many cluttered objects, that simulates settings such as very tight factory scenes. Cluttered scenes are missing in existing works that could be (mostly) solved with an “easy” solution that navigates around obstacles instead of in-between them. This easy solution side steps one of the harder challenges of the NMP problem as it doesn’t require an in-depth geometrical understanding of the scene (see the Related work section for more elaborate description of current works limitations). Finally, section 6 (limitations) discusses the gaps required to overcome in order to successfully use simulated NMP policies in the real world.

2. We will include *additional investigations to visual encodings*, in order to better understand what embeddings are being learned:
* Investigating how small perturbations in the parameters of the scene affect the embedding vectors.
* Predicting from the embedding vectors quantities such as the number of objects in the scene.

---

### Author Response · Authors · 2024-05-12
**Submission status request**

Dear reviewers and AC,

We kindly request an update on the status of this submission. We fully appreciate the time and effort you invest in the review process, and we're grateful for your dedication to the evaluation process.

Thank you very much for your time and consideration.

---

### Decision · Action_Editor_Uwas · 2024-05-21

**Recommendation:** Reject

**Comment:**

This paper reframes the task of robotic motion planning as a machine learning problem, where the robot learns the optimal actions to reach a goal while avoiding obstacles. It introduces a new benchmark environment called RoboArm-NMP and investigates how different learning approaches and design choices affect the robot's performance.

The study evaluates the robot's ability to generalize its learning to new scenarios, finding that it adapts better to new goals within a familiar environment than to entirely new obstacle configurations.

However, the paper's findings faced criticism during peer review. Concerns were raised regarding the rigor of the experiments, lack of clarity in the methodology, and insufficient evidence to support the claims. Reviewers also suggested a more comprehensive review of existing literature and better justification for using the proposed benchmarks.

Additional concerns were voiced about the need to better isolate the cause of the robot's generalization difficulties, the clarity of the experimental procedures, the appropriate interpretation of statistical significance, and the impact of the robot's action space on its performance.

Review requested changes were not addressed in a revised manuscript.

**Audience:**

Yes, improved motion planning remains a subject of interest in Robotics and RL Safety communities. As such, having a new benchmark would interest some individuals in TMLRs audience.

**Claims And Evidence:**

This paper casts motion planning amidst obstacles as a policy learning problem, presents a new benchmark called RoboArm-NMP, and explores how modeling choices (BC vs RL; goal representation; highsight experience replay etc) impact performance. Across easy/medium/hard benchmarks  involving getting an arm to reach a goal without colliding, with variable number of random obstacles and shapes in the scene, it concludes that generalization is more effective for unseen goals than unseen obstacle configurations.

The claims were somewhat contested in the reviews which asked for more rigor and insight, e.g., to disentangle whether generalization difficulties are due to vision encoders or policy overfitting; clarification of experimental procedure; scoping the claims appropriately based on degree of statistical significance; clarifying impact of action spaces etc. A separate set of concerns were raised around improving literature review and positioning of this work in that context, and providing greater motivation for adoption of the proposed benchmarks.

**Resubmission Of Major Revision:**

The authors may consider submitting a major revision at a later time.